# Traditional Medicinal Plants as a Source of Inspiration for Osteosarcoma Therapy

**DOI:** 10.3390/molecules27155008

**Published:** 2022-08-06

**Authors:** Liliya Kazantseva, José Becerra, Leonor Santos-Ruiz

**Affiliations:** 1Instituto de Investigación Biomédica de Málaga y Plataforma en Nanomedicina-IBIMA Plataforma BIONAND, 29590 Málaga, Spain; 2Centro de Investigación Biomédica en Red de Bioingeniería, Biomateriales y Nanomedicina (CIBER-BBN), Instituto de Salud Carlos III, 28029 Madrid, Spain; 3Department of Cell Biology, Genetics and Physiology, Universidad de Málaga, 29071 Málaga, Spain

**Keywords:** osteosarcoma, natural products, traditional medicinal plants, drug discovery, signaling pathway, combination therapy

## Abstract

Osteosarcoma is one of the most common types of bone cancers among paediatric patients. Despite the advances made in surgery, chemo-, and radiotherapy, the mortality rate of metastatic osteosarcoma remains unchangeably high. The standard drug combination used to treat this bone cancer has remained the same for the last 20 years, and it produces many dangerous side effects. Through history, from ancient to modern times, nature has been a remarkable source of chemical diversity, used to alleviate human disease. The application of modern scientific technology to the study of natural products has identified many specific molecules with anti-cancer properties. This review describes the latest discovered anti-cancer compounds extracted from traditional medicinal plants, with a focus on osteosarcoma research, and on their cellular and molecular mechanisms of action. The presented compounds have proven to kill osteosarcoma cells by interfering with different pathways: apoptosis induction, stimulation of autophagy, generation of reactive oxygen species, etc. This wide variety of cellular targets confer natural products the potential to be used as chemotherapeutic drugs, and also the ability to act as sensitizers in drug combination treatments. The major hindrance for these molecules is low bioavailability. A problem that may be solved by chemical modification or nano-encapsulation.

## 1. Introduction

Osteosarcoma, a bone cancer mainly arising in children and adolescents between the ages of 10 and 14, represents 3–5% of childhood cancer [1,2]. The annual incidence of this disease is 5.6 cases per million of paediatric patients [3]. Osteosarcoma occurs in the metaphysis of the wide portion of the long bones, which is characterized by an accelerated cell division, necessary for bone elongation [2]. During this process, cells can suffer different changes, such as loss of the tumour suppressor gene functionality, which will make them develop into a cancer. Moreover, some conditions are well known to predispose paediatric patients to osteosarcoma. These include retinoblastoma, Li–Fraumeni, and Rothmund–Thomson syndromes [4].

Even though this type of bone cancer is predominant in the young population, adults over the age of 50 are the second-highest risk group for suffering osteosarcoma [5]. In this case it is a secondary tumour caused by irradiation exposure to treat another type of cancer, which occurred previously in life [2]. Also, osteosarcoma can result from a sarcomatous transformation, a rare complication observed in elderly patients with Paget’s disease of the bone. In this group of patients, long bones are no longer the principal site affected by the tumour. Instead, jaw and pelvis are the most affected [6,7].

The current treatment option for osteosarcoma patients consists of surgical removal of the tumour and multi-agent chemotherapy, usually with methotrexate, doxorubicin, and cisplatin (Table 1). This therapy has enabled a five-year survival rate of 70% of patients with localized disease. However, the acquired chemoresistance observed in the metastasis or relapsed tumours is associated with a poor prognosis, with only a five-year survival rate of 20% [4,5]. Osteosarcoma chemotherapy has not substantially altered for decades, and the survival rate of patients with metastasis has remained unchanged for the last 20 years [5]. For this reason, the efficiency of osteosarcoma therapy needs to be urgently improved. One of the strategies is to discover novel anti-tumor drugs. Notwithstanding the potential of synthetic biology, this task has been more successfully achieved by taking advantage of nature’s molecular designs, a resource that has not yet been totally explored. New anti-tumour drugs could be found by looking at all that surrounds us, as nature is a tremendous source of chemical diversity [8,9]. Also, discovery of the potential application of the natural products has been facilitated with the contribution of modern computational techniques. Creation of databases with the aid of chemo-informatics enables researchers to access to the structural properties of the bioactive molecules [10]. Artificial intelligence and machine learning have been demonstrated to be useful in candidate drug structure design, as well as prediction of its targets, bioactivity, and toxicity [11].

Natural products encompass compounds derived from plants, fungi, and bacteria [18]. A first-in-class analysis between 1999 and 2013 demonstrated that 28% of the drugs approved by the US Food and Drug Administration (FDA) were natural products or their derivatives [9]. Indeed, current treatment of osteosarcoma relies on several compounds originating from natural sources. One such is doxorubicin, which is part of a standard drug combination given together with high-dose methotrexate and cisplatin. This chemotherapeutic drug is an anthracycline isolated from the bacteria *Streptomyces peucetius* [13,14]. Another one, etoposide, a semisynthetic compound derived from the mandrake plant *Podophyllum peltatum*, is typically used in combination with ifosfamide for osteosarcoma therapy [19,20].

To date, native people of the developing countries rely on the use of medicinal herbs to manage their healthcare issues [21]. For these people this kind of practice represents a value for its tradition and long-term belief of its effectiveness. In rural places it is difficult to access modern health facilities because of its costs, making it more likely in these areas to seek solutions from traditional healers. As a result, there is an expanded use of herbs for illnesses that range from minor diseases to serious ones, such as cancer [22,23,24]. In Sri Lanka, ginger, *Zingiber officinale*, is employed to treat oesophageal, liver, and gastrointestinal cancer [25]. The leaves of the Black Calla Lily, *Arum palaestinum* Boiss, are commonly used as a herbal remedy for cancer patients in Pakistan [26]. Many studies in medicinal plants extracts confirmed their anti-tumor capacities [27]. Herb preparations represent a potential source for anti-cancer agents that still should be explored. For this reason, the aim of this review is to present compounds derived from medicinal plants currently in research for osteosarcoma treatment.

## 2. Oridonin

Oridonin is a diterpenoid isolated from the Isodon plant, *Rabdosia rubescens,* in 1970 (Figure 1). This medicinal herb has been traditionally used by native people in China to alleviate pain, manage inflammation, and treat oesophageal cancer [28,29]. Several studies have demonstrated the anti-tumour capacities of oridonin, revealing a wide spectrum of pharmacological activities that include: induction of cell autophagy and apoptosis; arrest of cell-cycle progression; and inhibition of angiogenesis [28,30,31,32,33,34]. In the case of osteosarcoma, the interest in this natural product is recent. There are few available reports, but with promising results. Oridonin exerts its anti-cancer activity by inducing mitochondria-mediated apoptosis through augmentation of Bax/Bcl-2 ratio, and activation of caspase–3 and 9, accompanied by augmented reactive oxygen species (ROS) production. Moreover, disruption of several signalling pathways promotes apoptosis. For example, the natural product is able to activate p38 MAPK, JNK, and PPAR-γ, while it inhibits Akt and Nrf2 pathways [35,36,37].

Chemoresistance observed in metastatic cancer is a challenge for a successful therapy. In a study by Wang and Shu [31], oridonin showed the ability to suppress metastasis of human ovarian cancer by blocking mTOR signalling pathway and increasing the expression of the downstream gene FOXP3. To confirm and clarify these effects of the natural product on osteosarcoma cells Sun et al. [38] focused on the epithelial-to-mesenchymal transition (EMT), a process where cells lose their epithelial cell–cell adhesion pattern and acquire a mesenchymal phenotype, with migratory and invasive properties. During EMT the E–cadherin marker of epithelial cells disappears and instead N–cadherin, a mesenchymal marker, is expressed. Oridonin was able to prevent this process by increasing the transcription of the E–cadherin, while N–cadherin was down-regulated. Besides, TGF-β1, a key receptor involved in the EMT initiation, was inhibited by preventing the activation of the downstream Smad2 and Smad3, necessary for the expression of a variety of genes involved in cancer progression [38]. These results demonstrate the potential of oridonin to be an effective candidate for advanced stages of osteosarcoma, where a poor prognosis still remains.

As with other chemotherapeutic drugs, an important drawback of natural products as anti-cancer drugs is the risk of resistance that can appear within months. For this reason, to increase the efficiency of osteosarcoma treatment, researchers rely on combination therapy. Recently, a synergistic cytotoxic effect has been described for oridonin and doxorubicin in an in vitro model of osteosarcoma [39]. Wang et al. [28] investigated the possibility of combining the natural product with Nutlin–3, an inhibitor of mouse double minute 2 (Mdm2) protein, a negative regulator of p53 function. The combination was able to inhibit cell viability in those osteosarcoma cells that were bearing a wild-type p53. In osteosarcoma in vitro models with compromised p53 function, such as MNNG/HOS cells, with mutated p53, or Saos–2 cells, with null p53, a reduction in cell viability was detected, but the calculated combination index (CI) showed there was no synergistic activity between oridonin and Nutlin–3 [28]. The administration of both drugs represents a novel therapeutic strategy, although further studies are needed using different doses of oridonin and Nutlin–3, as synergistic and antagonistic effects depend on the dosage.

Despite the promising efficacy of oridonin for osteosarcoma therapy, its clinical application is hindered by its limited water solubility and bioavailability. To overcome such drawbacks oridonin derivatives have been synthesised, such as geridonin and CYD0618. Having the natural product come into osteosarcoma study recently, there are no reports evaluating its analogs, but they have been studied for other types of cancer [40,41,42].

## 3. Wogonin

Wogonin is one of the active components extracted from the root of Baikal skullcap, *Scutellaria baicalensis* Georgi, commonly used for the treatment of inflammatory diseases in China (Figure 2) [43]. It was shown to have anti-cancer properties in osteosarcoma U2OS cells through the activation of apoptosis. Wogonin triggered this process by inducing ROS production, responsible of mitochondrial membrane potential disruption, release of additional ROS, and caspase–3 activation [44].

Cancer is a complex disease where tumour cells are not the unique component of the tumour microenvironment. Endothelial cells, tumour-associated macrophages (TAM), carcinoma-associated fibroblasts, and cancer stem cells (CSC) are other important elements that can contribute to cancer metastasis and relapse [45]. Importantly, for osteosarcoma therapy wogonin demonstrated to have inhibitory properties on metastatic tumour. In CSC the natural product favoured ROS generation by reducing the expression of the antioxidant peroxiredoxin 5 (PRX5) and inhibiting the transcriptional factor STAT3 [46]. Moreover, wogonin was able to affect the expression of matrix metalloproteinase 9 (MMP–9) in osteosarcoma stem cells, a protein involved in the degradation of the extracellular matrix (ECM) and promoter of the angiogenesis necessary for malignant cells invasion. In this way, the renewal capacity of CSC was inhibited, diminishing the potential risk of these cells to contribute with further cancer cells necessary for metastasis [47].

The Implication of TAM in the anti-tumour effect of wogonin was observed in a mice model bearing a highly metastatic osteosarcoma: LM8 cells. The inhibition of cyclooxygenase–2 (COX–2) expression and IL–β production in activated TAM led to the suppression of vascular endothelial growth factor (VEGF)-induced lymphangiogenesis [43]. The ability of wogonin to target different components of the tumour microenvironment makes it a valuable candidate for osteosarcoma therapy that should receive further attention.

## 4. Oleuropein

Oleuropein is a secoiridoid, one of the most abundant components found in the fruits and leaves of the olive tree *Olea europaea* L. (Figure 3). The olive oil is a well-known source of energy, and important part of the Mediterranean diet [48]. Besides, the leaves, fruit, bark, and oil of the olive tree have been used in traditional medicine among different countries. In the Mediterranean folk medicine, olive leaves preparations are commonly used against gout, while in Tunisian folk medicine, they are used to treat inflammation and bacterial infections, such as gingivitis or otitis [49,50]. In some regions of Iran, the leaves are used as a remedy for muscle and joint pain [51]. A number of studies have demonstrated the anti-tumor effect of oleuropein in different types of cancer [52,53,54]. In the case of osteosarcoma, it has shown anti-proliferative properties, where autophagy was implied. The role of autophagy induction by oleuropein in cancer cells is still to be elucidated, as this process has a dual role, being in some cases promoter and in another suppressor of the tumour growth [48,55,56]. This is an important information to take into account, as it will determine the success of the therapy.

The available studies about oleuropein in osteosarcoma employ the drug to improve the efficiency of other, better-known drugs. Co-administration of doxorubicin with oleuropein in an in vitro model of osteosarcoma resulted in enhanced inhibition of osteosarcoma cell proliferation. Given the cumulative, dose-dependent, side effects of doxorubicin (such as cardiotoxicity), combination therapies that allow reducing its dosage are actively sought [48]. In another study, synergism between oleuropein and 2–methoxyestradiol (2–ME), a metabolite of 17β–estradiol with anti-cancer properties, was observed [56].

## 5. Evodiamine

Evodiamine is an alkaloid isolated from the fruit of the Chinese medicinal plant *Evodia rutaecarpa* (Figure 4). Traditionally, this herb has been used to treat inflammation diseases, abdominal pain, headache, dysentery, and postpartum haemorrhage [57,58]. Research studies demonstrated evodiamine to possess anti-tumor properties by inhibiting the proliferation of several types of cancer, including osteosarcoma [59,60,61,62,63]. In bone cancer these effects were caused in a time- and dose-dependent manner. Evodiamine exerted its anti-cancer activities through mitochondrial apoptosis induction, evident from increased levels of Bax, caspase–3 and PARP, while the expression of Bcl–2 and Survivin were decreased. Cell-cycle arrest at the G2/M or G0/G1 phase was also implicated. Further studies on the mechanisms of the anti-cancer effects of evodiamine found inactivation of PTEN/PI3K/Akt and inhibition of Wnt/β–catenin signalling pathways [59,60,61].

Osteosarcoma is a tumour that has a tendency of invasion and metastases in lungs. These factors are responsible for the poor prognosis in the advanced stages of bone cancer [2,4]. Evodiamine was shown to be effective in suppressing the EMT process associated to metastatic processes. Thus, the natural product increased the expression in E–cadherin, while N–cadherin, Vimentin, MMPs, and the transcription factor Snail were down-regulated [59].

The efficacy of Evodiamine as a potential candidate for osteosarcoma treatment has been studied in vivo in a xenograft model, where 143B cells were subcutaneously injected into the backs of athymic nude mice. After the treatment with the natural product, tumour growth was suppressed in a dose-dependent manner and the expression of the proliferating cell nuclear factor (PCNA) in tumour cells was decreased, thus confirming the inhibition of tumorigenesis [61]. Based on these findings, Evodiamine is another candidate to be further evaluated in osteosarcoma treatment.

## 6. Parthenolide

Parthenolide is a sesquiterpene lactone found in the feverfew plant, *Tanacetum parthenium* (Figure 5). This herb has been a traditional folk medicine across Europe, used to treat insect bites, psoriasis, toothache, rheumatoid arthritis, migraine, and fever [64,65]. Nowadays, feverfew supplements are given for migraine headaches, in the form of capsules or tables of dried leaves, standardized to contain 0.2–0.4% of parthenolide [64]. Anti-cancer action of this natural product has been found against different types of cancer, such as breast, colon, prostate, and skin [66,67,68,69].

NF–κB is a dimeric transcription factor that is up-regulated in osteosarcoma [70,71,72]. The cJun N–terminal kinase (JNK) is a signalling pathway involved in apoptosis induction upon cytoplasmic stress or DNA damage [73]. In osteosarcoma, activation of NF–κB and inhibition of JNK pathway favours malignant cell survival, thus affecting the outcome of the cancer therapy. Parthenoline was reported to suppress NF–κB activity and prompt the activation of JNK in a dose-dependent manner, followed by caspase-independent bone cancer cell death [74,75]. One of the proposed mechanisms of the natural product is through ROS generation, which leads to dissipation of mitochondrial membrane potential. As a result, there is a release of cytochrome c and apoptosis-inducing factor (AIF), followed by nuclear translocation of AIF, which causes chromatin condensation and fragmentation [75]. Another mechanism by which parthenolide causes cell death is through autophagy induction mediated by ROS production [74]. Furthermore, the natural product showed metastasis preventive effects as it inhibited the development of lung metastasis of LM8, a highly metastatic subclone of Dunn murine osteosarcoma cell line [72].

In some cases, the localization of osteosarcoma tumours, such as pelvis or jaw, makes it difficult to perform its surgical resection, as these regions are surrounded by vital organs. For this reason, post-operative radiation therapy is usually given to kill the remaining cancer cells. Moreover, radiation is used to relieve the pain for unbearable bone metastasis and prolong the survival of the patient. Combination of radiotherapy with chemotherapy has been observed to improve the effectiveness of the cancer treatment, in general [76,77]. However, osteosarcoma is known for its radioresistance. For these reason, radiosensitization strategies followed by a posterior chemotherapy are under investigation. It is established that the expression of NF–κB contributes to osteosarcoma radioresistance [70,78]. Zuch et al. [63] found enhanced cell death of an aggressive derivative of Saos–2, LM7, when parthenolide and ionizing radiation were given together. In this case, treatment with the natural product reduced NF–κB activity and increased oxidative stress. The posterior irradiation further diminished osteosarcoma cell viability [69]. Similar findings were detected in LM8 transfected with a reporter construct of NF–κB, Luc–LM8, in vitro and in vivo [78]. Importantly, synergy between parthenolide and ionizing radiation was observed against the cancer stem cells subpopulation of LM7 [69]. These data make parthenolide an attractive sensitizing agent of osteosarcoma cells prior to radiotherapy. Finally, the effects of parthenolide seem to be specific to cancer cells, being nontoxic to healthy cells, according to Zuch et al. [69] where cell viability of LM7 and human fetal osteoblast hFOB 1.19 was assessed using LIVE/DEAD Viability/Cytotoxicity kit [69].

## 7. Shikonin

Shikonin is a naphthoquinone isolated from the Chinese medicinal plant *Lithospermum erythrorhizon*, widely used for treating inflammation and wound healing (Figure 6) [79,80]. It was shown to reduce osteosarcoma cell viability in a time- and dose-dependent manner [79,81]. Moreover, shikonin can suppress osteosarcoma invasiveness through MMP13 inhibition [82,83]. The natural product exerts its anti-cancer activity through different molecular mechanisms that vary according to the cell line and treatment time. In 143B cells, increasing levels of ROS production and activation of extracellular signal-regulated kinase (ERK), in response to shikonin exposure, caused apoptosis [79]. On the other hand, K7, K12, K7M3, and U2OS cell death after shikonin treatment occurred through necroptosis induction, as the level of proteins RIP1 and RIP3, involved in this process, were increased. These effects were reversed when necrostatin–1, a specific inhibitor of necroptosis, was used, while no protective effects were observed with the addition of the general caspase inhibitor, Z–VAD–FMK [81].

Interestingly, shikonin has also proved to be a promising compound for combination treatment. The natural product acted in synergy with low doses of doxorubicin and further enhanced the apoptotic effect of this drug in osteosarcoma cells on account of caspase–3 and caspase–8-dependent apoptotic pathway [84]. The ability of shikonin to potentiate the effects of low doses of doxorubicin should be taken into account since it could reduce the previously mentioned side-effects of doxorubicin, such as cardiotoxicity, that is known to produce a heart failure in survivors of paediatric cancer later in life [85].

The observed anti-cancer effects of shikonin in vitro were confirmed in vivo in an orthotopic osteosarcoma model, where the K7 osteosarcoma cell line was injected into the medullary cavity of mice tibia. After tumours developed, the mice were treated with shikonin for two weeks, after which they were euthanized and the tumours analysed. Tumour size in the shikonin-treatment group was significantly smaller, as compared to the untreated control. Necroptosis induction was deduced to have caused tumour cell death in vivo, as RIP1 and RIP3 were significantly increased in the primary tumour tissue. In a parallel experiment performed to asses metastasis of osteosarcoma cells to the lungs, shikonin reduced the extent of lung metastasis and visibly prolonged the survival of mice in this experimental group, as compared with the untreated control group [81]. Based on these findings the natural product could be a propitious candidate to prevent metastasis in osteosarcoma patients, as well as to improve osteosarcoma chemotherapy.

In order to boost the therapeutic efficiency of shikonin, Kong et al. [80] synthetized different derivatives, acylated by distinct fluorinated carboxylic acids at the side chain, and evaluated their anti-cancer activity in MG63 osteosarcoma cell line. The authors found that, from 11 synthesised compounds, the one named S7 presented a strong anti-cancer activity against MG63. In addition, docking simulations showed S7 to act as an inhibitor of tubulin polymerization [80].

## 8. Berberine

Berberine is an isoquinoline alkaloid found in different types of plants, such as *Berberis vulgaris*, *Berberis aristata*, *Berberis aquifolium*, *Coptis chinensis,* and *Hydrastis canadensis* (Figure 7) [86]. In traditional Chinese medicine it has been used for its anti-inflammatory and anti-microbial properties [86,87]. Numerous studies reported berberine to be effective against lymphoblastic leukemia, colorectal, prostate, breast, and esophageal cancer [88,89,90,91,92]. Similarly, it inhibited proliferation, viability, migration, and colony formation of osteosarcoma cells in a time- and dose-dependent manner [87,93]. There are several proposed underlying mechanisms that explain berberine’s anti-tumour effects. One of them involves genomic damage triggering p53 activation, with subsequent induction of cell-cycle arrest and apoptosis [93]. Zhu et al. [94] also found berberine to produce both apoptosis and DNA double strand breaks. In another study, the natural product was able to reduce the expression of caspase–1, a cysteine protease involved in the activation of proinflammatory cytokines, and its downstream target IL–1β, leading to osteosarcoma cell growth inhibition. This finding was confirmed in vivo in a xenograft mouse model [95].

The success of cancer metastatic dissemination throughout the body depends on its ability to migrate and degrade the extracellular matrix. Berberine has demonstrated to be effective against metastatic osteosarcoma by inhibiting migration and colony formation of these cells. This was achieved through the natural product’s ability to suppress the activity of MMP2. Moreover, berberine could also inhibit EMT, as suggested from the observed increased expression of E–cadherin and reduced expression of N–cadherin, vimentin, and fibronectin observed in berberine-treated MG63 osteosarcoma cells in vitro [87].

Cisplatin is one of the chemotherapeutic drugs given along with doxorubicin and high-dose methotrexate for osteosarcoma treatment [5]. Its applicability is limited due to cancer cells resistance and its serious side effect, i.e., nephrotoxicity. As it occurs with doxorubicin, novel adjuvant drugs are being sought that can be combined with cisplatin in order to achieve a safer treatment, with lower cisplatin dosing but equal therapeutic effect. Experimental evidence supports that berberine potentiates the anti-cancer activity of cisplatin. In an in vitro model of osteosarcoma (MG63 cells), the combination of cisplatin and berberine significantly inhibited cell migration and invasion to a higher extent than individual drug administration. These anti-tumour effects were attained by cell-cycle arrest in G_0_/G_1_ phase and apoptosis induction through MAPK signalling pathway repression [96].

Berberine represents a potential candidate for osteosarcoma therapy. However, further studies are needed to elucidate the safeness of the natural product as its ability to produce DNA damage is a concern requiring attention.

## 9. Triptolide

Triptolide is a diterpenoid epoxide obtained from the Thunder God Vine, *Trypterygium wilfordii*, a medicinal plant used for centuries in China to treat inflammatory and autoimmune diseases (Figure 8) [97,98]. Several studies have shown the ability of this natural product to reduce proliferation and cause cell death in pancreatic cancer, non-small cell lung cancer, and hepatocellular carcinoma [99,100,101]. In the case of osteosarcoma, triptolide has shown to exert diverse effects on apoptosis, autophagy, and angiogenesis [102,103,104]. It has been observed that the natural product can up-regulate Fas death receptor and its ligand FasL, and also increase the levels of caspase–3, –8, –9 and enhance cytochrome c release, pointing triptolide-induced apoptosis to be caused through activation of both extrinsic (FasL, Fas, caspase–8) and intrinsic (caspase–9 and cytochrome c) mitochondrial pathways [102]. Similarly, Zhao et al. [97] proposed the anti-tumour effects of triptolide to occur via the activation of DR–5/p53/Bax/caspase–9/–3 and DR–5/FADD/caspase–8/lysosomal/cathepsin B/caspase–3 signalling pathways.

The variety of mechanisms of action proposed for triptolide make this drug a possible multifunctional compound against osteosarcoma. Another explored pathway suggests that triptolide reduces osteosarcoma cell viability by decreasing the expression of mitogen-activated protein kinase phosphatase 1 (MKP–1), a repressor of MAPK signalling pathway, and that of heat shock protein 70 (Hsp70), a chaperone whose increased levels are speculated to contribute in chemoresistance [105]. Besides, the anti-cancer effects of triptolide were reported to occur through autophagy induction by inactivating Wnt/β–catenin signalling. In addition, it may suppress angiogenesis in osteosarcoma cells [103]. A dual-specificity protein phosphatase 1 (DUSP1) that functions as a negative regulator of MAPK family members, and is known to be up-regulated in bone cancer, is another target proposed for the natural product. DUSP1 inhibition enables the activation of ERK1/2 and JNK1/2, leading to further induction of apoptosis [104].

Combination therapy is another strategy through which triptolide could contribute to improving current osteosarcoma therapies. In several studies, it has been observed that the natural product enhances the sensitivity of cancer cells to doxorubicin and cisplatin [104,106]. On other hand, triptolide co-administration with AMD3100, an antagonist of C–X–C chemokine receptor type 4 (CXCR4), induced apoptosis, and inhibited proliferation and invasion of osteosarcoma cells in vitro. In vivo, this combination suppressed primary tumour growth and lung metastasis, thus representing a promising strategy against metastatic bone cancer [107]. However, the low solubility of triptolide in aqueous solutions limits, at present, its application in in vivo models. For this reason, minnelide, a prodrug of the natural product, has been developed. Similar to triptolide, minnelide has shown to be an effective compound with the ability to reduce tumour growth and metastasis in nude mice. Importantly, both minnelide and triptolide have been able to negatively affect osteosarcoma cells, while having minimal repercussion on osteoblastic cells, pointing these drugs to be safe for chemotherapeutic usages [108].

## 10. Novel Natural Products

Recently, novel natural products have been evaluated for the first time for osteosarcoma therapy. Phillygenin is a lignan component isolated from the dried fruit of *Forsythia suspense*, a traditional Chinese medicinal plant used for heat clearing and swelling reduction [109]. Previously it had shown anti-tumour effects against a non-small cell lung cancer [110]. In osteosarcoma, phillygenin decreased cell growth and inhibited cell migration of 143B, HOS, and SJSA by interfering with STAT3 signalling pathway [111]. Another novel compound is oxyresveratrol, extracted from *Cortex mori*, whose roots have been used in traditional Chinese medicine to treat asthma, cough, and water swelling [112]. In modern research it showed to be effective against neuroblastoma, breast cancer, and hepatocellular carcinoma [113,114,115]. Also, oxyresveratrol revealed promising results in osteosarcoma treatment. The natural product inhibited Saos–2 cell viability and induced apoptosis by reducing the phosphorylation level of STAT3 [116]. However, further studies of the underlying mechanisms of phillygenin and oxyresveratrol are still needed to elucidate the applicability of these novel compounds for osteosarcoma therapy. Calonghi and coworkers investigated the effect of the lipophilic fraction of two *Paeonia* species on different types of cancer cells. They found that ovarian carcinoma and osteosarcoma cells were sensitive to the extracts, which caused alteration of the mitochondria membrane potential and production of ROS. Interestingly, paeonol, the active compound characteristic of peonies, was not present in the studied fraction. This suggests the presence of alternative, not-yet-identified, anti-cancer compounds in *Paeonia* plants [117].

The summary of the effects of the natural products on osteosarcoma discussed in the present review are outlined in Table 2.

## 11. Discussion and Future Perspectives

Osteosarcoma is a cancer type for which, despite abundant research, no significant therapeutic improvements have been achieved in the last decades. However, if something can be learnt from research, it is that osteosarcoma is a very variable cancer type, in both its clinical manifestation and its cellular and molecular basis. Gene sequencing, molecular profiling, and phenotypic screening of osteosarcoma biopsies and cell lines have shown that the aberrant behaviour of osteosarcoma cells can have its origin in a wide range of genetic, epigenetic, and molecular changes [118]. Consequently, it is unlikely there will ever be a “one-fits-all” chemotherapeutic regime for this disease. A precision medicine approach, in which different drug combinations will be suggested by the phenotypical profiling of each patient’s tumour, is far more plausible. Also, because of the complexity of osteosarcoma, these treatments are expected to combine different drugs with different targets, including, for example, intracellular pathways, microenvironmental signalling, or cancer stem cells. Therefore, the fight against osteosarcoma is expected to require a broad spectrum of chemotherapeutical agents to be combined ad hoc for each patient. Paradoxically, at present only four drugs are used for this cancer (cisplatin, doxorubicin, ifosfamide, and methotrexate), three of them sharing DNA as their common target. Hence the urgent need for new drugs to be used in combination therapies that attack the tumour in its multiple metabolic and cell-cycle checkpoints aberrations.

This review is focused on the most recently identified compounds from medicinal plants that can have a potential for osteosarcoma treatment. These compounds have been found able to inhibit cell growth or kill osteosarcoma cells by acting on different pathways, which included apoptosis and autophagy induction, ROS generation, EMT inhibition, etc.

When it comes to designing a drug regime for osteosarcoma, the heterogeneity of the tumour tissue is of special concern. It contains neoplastic cells and cancer stem cells, associated to a tumour-associated stroma, vasculature, and immune infiltrate, all of them conforming a niche that governs tumour characteristics, such as invasion, metastasis, and dormancy [119]. Among the compounds discussed in this review, wogonin seems to be able to act against different cell components of the tumour microenvironment, including cancer stem cells. Different studies have described that it can reduce migration and self-renewal tumour ability, and exert anti-angiogenic effects [43,44,46,47], making it a promising drug to avoid recurrence or metastatic spread.

Interestingly, several of the revised natural products, including oridonin, oleuropein, shikonin, and triptolide, have proven to work in synergy with doxorubicin, a drug commonly used in the current treatment of osteosarcoma. The combination with these drugs helped to reduce doxorubicin dosage, thus limiting its very serious side-effects. Chemotherapeutic regimes based on the combination of novel drugs with classical ones are worth exploring, given that chemotherapy-induced toxicity is an unresolved issue for osteosarcoma, as well as other cancers [120,121]. Of similar interest would be the combination therapy of some of the herein described plant-derived products with other synthetic drugs currently undergoing clinical trials. Thus, oridonin, oleuropein, and triptolide have been found to present a synergistic effect with Nutlin–3, 2–methoxyestradiol, and AMD3100, respectively, when used against osteosarcoma [28,56,107].

In order to futureproof natural products in drug development for bone cancer there are still gaps that should be filled with additional investigation. Most of the articles describing the effects of new compounds on osteosarcoma use only one cell line. As mentioned, osteosarcoma is a complex, heterogenic cancer, where neoplastic cells differ in their phenotype and mutation status, leading to a variable response (sensitivity or resistance) to drugs. Studies that include, from its most early stages, different bone cancer cell lines are interesting in order to capture this diversity. Another key for accelerating osteosarcoma drug discovery would be the introduction of 3D cultures at early stages of in vitro research. Three-dimensional cultures are better at mimicking the complex signalling within the tumour microenvironment, and are, therefore, better at predicting the in vivo response of the tumour to a drug regime [122,123].

A handicap frequently faced by plant-derived compounds in in vivo assays and clinical trials is their low bioavailability, which can be due to several causes, such as instability in low pH environments, poor intestinal absorption, high rate of metabolism, or rapid systemic elimination [124]. Sometimes the drug has a low water solubility, hindering both oral and injected administration routes. This is, for example, the case with oridonin, that is being circumvented through the synthesis of different analogues [40,41,42]. Chemical modifications of the plant-derived bioactive compounds can confer them improved solubility and pharmacokinetic properties. For example, Calonghi and coworkers have recently tested a series of regioisomers of the hydroxystearic acid (HSA) on different cancer cell lines, finding that positional isomers possess distinct biological activities [125]. Although osteosarcoma cells were not included in this paper, it opens a new avenue for studying how chemical modification of known natural compounds can modulate their activity. Another alternative that is gaining ground is encapsulation in liposomes, micelles, or nanoparticles. These delivery systems can protect the drug on its way to the tumour, and also provide additional pharmacokinetic advantages, such as the ability to cross the cell membrane or the specific vectorization towards the tumour [126,127]. In the case of bone cancers, decorating the nanoparticles with bisphosphonates has proven to efficiently and selectively deliver nanoparticles to the bone [128].

## 12. Conclusions

This review has presented the latest plant-derived compounds to show promising activity against osteosarcoma. These products present a wide variety of targets, including intracellular pathways and bone cancer microenvironment signalling. They have potential to be used as chemotherapeutic agents due to their innate anti-tumour effect, or as sensitizers in multi-drug regimens.

The natural world is an inexhaustible source of bioactive compounds and, notwithstanding huge achievements in synthetic drug manufacturing, nature, particularly plants, keeps being the main supplier of pharmaceuticals. Although more preclinical research and clinical trials are still needed, the positive results found to date with plant-derived drugs encourage searching for new compounds in the still widely unexplored natural world. Research into the effects of these products should go hand-in-hand with research into the cellular and molecular causes of osteosarcoma, in order to design safe, efficient, and precise treatments for this malady.

## Figures and Tables

**Figure 1 molecules-27-05008-f001:**
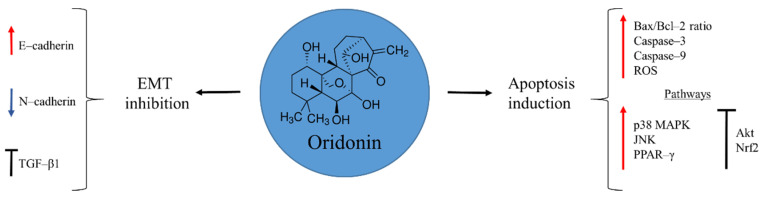
Oridonin in osteosarcoma inhibition.

**Figure 2 molecules-27-05008-f002:**
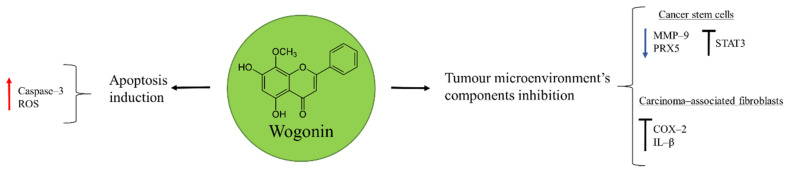
Wogonin in osteosarcoma inhibition.

**Figure 3 molecules-27-05008-f003:**
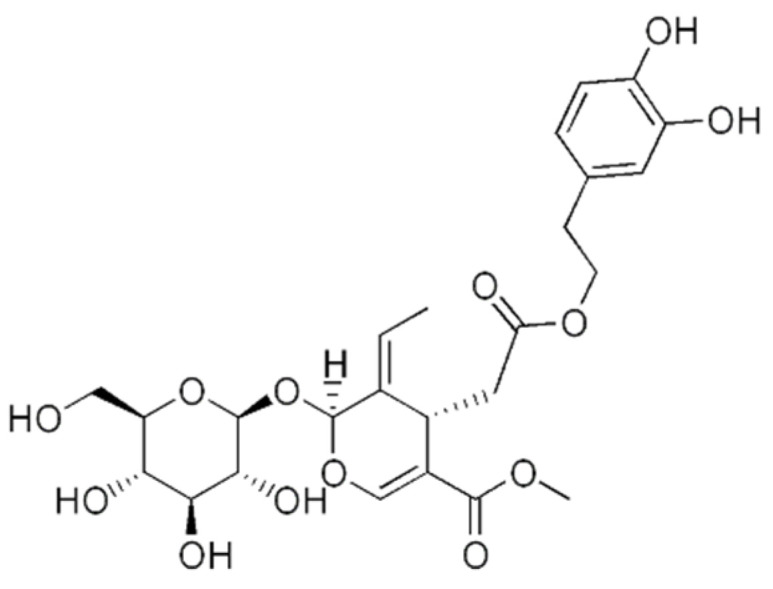
The chemical structure of oleuropein.

**Figure 4 molecules-27-05008-f004:**
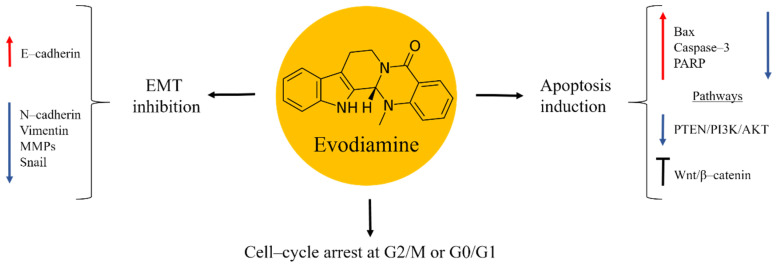
Evodiamine in osteosarcoma inhibition.

**Figure 5 molecules-27-05008-f005:**
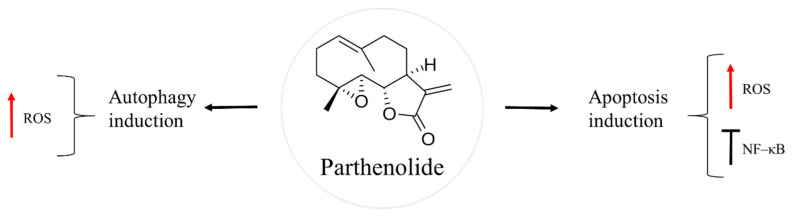
Parthenolide in osteosarcoma inhibition.

**Figure 6 molecules-27-05008-f006:**
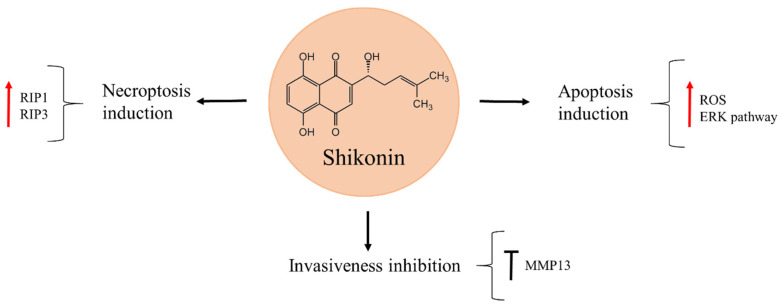
Shikonin in osteosarcoma inhibition.

**Figure 7 molecules-27-05008-f007:**
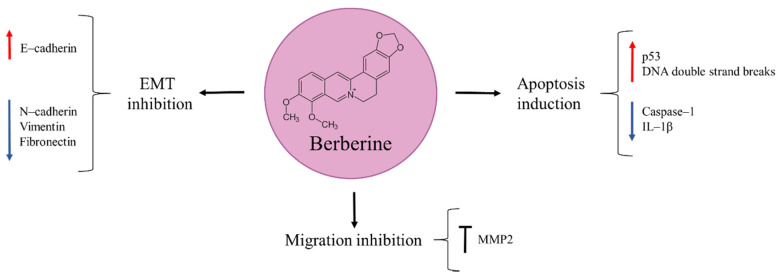
Berberine in osteosarcoma inhibition.

**Figure 8 molecules-27-05008-f008:**
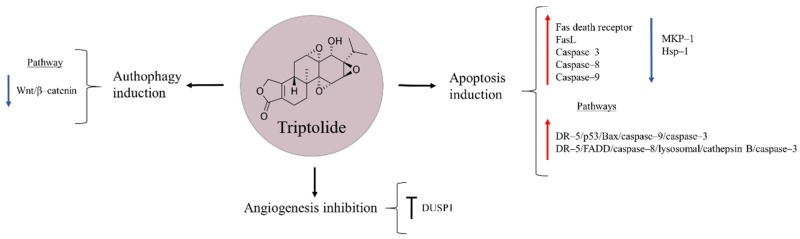
Triptolide in osteosarcoma inhibition.

**Table 1 molecules-27-05008-t001:** Chemotherapeutic drugs used in osteosarcoma therapy.

Chemotherapeutic Drug	Description	Effect	Reference
**High-dose methotrexate**	Folic acid analogue	Induced apoptosis and inhibited DNA synthesis, through blockage of dyhidrofolate reductase (DHFR)	[12]
**Doxorubicin**	Anthracycline	Induced cell death through intercalation between DNA strands, Topoisomerase II complex stabilization, and induction of oxidative stress	[13,14]
**Cisplatin**	Platinum-based compound	Induced apoptosis and inhibited genetic material replication and repair through DNA adduct formation	[15]
**Ifosfamide**	Alkylating a gent	Induced apoptosis and inhibited genetic material replication through DNA intra- and inter-strand crosslinks	[4,16,17]

**Table 2 molecules-27-05008-t002:** The effects of different natural products on osteosarcoma.

Natural Product	Dose	Target	Effect	Model	Cell Line	Reference
**Oridonin**	0–200 μM	MMP–2, 3, 9 and STAT3 pathway	Induced apoptosis, inhibited proliferation, migration, and invasion	In vitro	U2OS	[35]
**Oridonin**	0–100 μM	PPAR–γ and Nrf2 pathways	Induced apoptosis and inhibited proliferation	In vitro and in vivo	MG63 and HOS	[36]
**Oridonin**	0–100 μM	Akt, ERK, p38 MAPK and JNK pathways	Induced apoptosis and suppressed proliferation	In vitro	MG63, U2OS and Saos–2	[37]
**Oridonin**	0–4 μM; 0, 10 and 15 mg/kg	TGF-β1/Smad2/3	Inhibited EMT, migration, invasion, and lung metastasis	In vitro and in vivo	MG63, U2OS and 143B	[38]
**Wogonin**	0–100 μM; 25 and 50 mg/kg	-	Reduced tumour growth, metastasis, angiogenesis, lymphangiogenesis, and TAM number	In vitro and in vivo	LM8	[43]
**Wogonin**	0–150 μM	ROS and caspase–3	Induced apoptosis	In vitro	U2OS	[44]
**Wogonin**	0–80 μM	ROS	Reduced cell viability, proliferation, stemness, migration, and self-renewal capacities	In vitro	CD133+ Cal72	[46]
**Wogonin**	0–80 μM	MMP–9	Induced apoptosis, inhibited migration invasion, and reduced renewal capacities	In vitro	CD133+ Cal72	[47]
**Oleuropein**	50–400 μM	-	Reduced proliferation	In vitro	MG63 and Saos–2	[55]
**Evodiamine**	0–32 μM	Wnt/β–catenin pathway	Induced apoptosis, inhibited proliferation, migration, and invasion; suppressed EMT and caused cell-cycle arrest	In vitro	MG63 and 143B	[59]
**Evodiamine**	0–12.5 μg/mL	Bcl–2, Bax, caspase–3, and survivin	Inhibited proliferation and induced apoptosis	In vitro	U2OS	[60]
**Evodiamine**	0–4 μM	PTEN/PI3K/Akt pathway	Inhibited proliferation, induced apoptosis and caused cell-cycle arrest	In vitro and in vivo	143B	[61]
**Parthenolide**	0–25 μM	ROS	Induced cell death, autophagy, and mitophagy	In vitro	MG63 and Saos–2	[74]
**Parthenolide**	0–100 μM	AIF	Induced cell death	In vitro	MG63	[75]
**Parthenolide**	0–20 μM	NF–κB pathway	Induced cell death and radiosensitivity	In vitro	LM7	[70]
**Parthenolide**	0 and 1 μg/mL; 1 and 2 mg/kg	NF–κB	Enhanced radiosensitivity and inhibited tumour growth	In vitro and in vivo	LM8	[78]
**Shikonin**	0–8 μM	ROS, ERK, and Bcl–2	Induced apoptosis	In vitro	143B	[79]
**Shikonin**	0–15 μM; 2 mg/kg	RIP1 and RIP3	Induced cell death, necroptosis, and increased the survival time in metastatic disease	In vitro and in vivo	K7, K12, K7M3, U2OS and 143B	[81]
**Berberine**	0–80 μM	MMP–2, H3K27me3, and EZH2	Inhibited proliferation, migration and EMT	In vitro	MG63	[87]
**Berberine**	0–50 μg/mL	p53, p21, p27, and cyclin E	Induced apoptosis, inhibited proliferation, and caused cell-cycle arrest	In vitro	U2OS, Saos–2 and HOS	[93]
**Berberine**	0–80 μM	DNA	Induced DNA damage and apoptosis	In vitro	MG63	[94]
**Berberine**	0–120 μg/mL; 20 mg/kg	Caspase–1/IL–1β pathway	Induced apoptosis, inhibited tumour growth, and modulated inflammation in tumour microenvironment	In vitro and in vivo	MG63 and Saos–2	[95]
**Triptolide**	0–200 nM	DR–5/p53/Bax/caspase–9/–3 and DR–5/FADD/caspase– 8/lysosomal/cathepsin B/caspase–3 pathways	Suppressed cell viability and induced apoptosis	In vitro	MG63	[97]
**Triptolide**	0–500 nM	procaspase–8,–9, Bcl–2, Bid, Fas, FasL, Bax, caspase–3, PARP, mitochondrial and cytosolic cytochrome *c*	Inhibited cell growth, induced cell-cycle arrest, and apoptosis	In vitro	U2OS	[102]
**Triptolide**	0–200 nM	HIF–1alpha, VEGF, and Wnt/β–catenin pathway	Inhibited angiogenesis, induced apoptosis through autophagy activation	In vitro	MG63	[103]
**Triptolide**	0–400 nM; 0.2 mg/kg	DUSP1	Inhibited cell viability, migration and invasion; induced apoptosis and caused cell-cycle arrest	In vitro and in vivo	MG63, U2OS and UMR–106	[104]
**Phillygenin**	0–200 μM	SHP–1/JAK2/STAT3 pathway	Inhibited cell growth and motility	In vitro	143B, HOS, SJSA	[111]
**Oxyresveratrol**	0–45 μM	STAT3 pathway	Inhibited cell viability and induced apoptosis	In vitro	Saos–2	[116]

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
