# Peer review of "Traditional Medicinal Plants as a Source of Inspiration for Osteosarcoma Therapy"

_molecules, 2022, doi:10.3390/molecules27155008_

Round 1

Reviewer 1 Report

This review article discusses Traditional medicinal plants as a source of inspiration for osteosarcoma therapy. The authors also present our contribution to bone cancers among pediatric patients. Before recommending this article for publication, there are some shortcomings that should be resolved.

General comments

Overall the experiment is well designed and presented in a good way but the English of the whole manuscript should be revised. Long sentences should be avoided.

Abstract

This section is well written but the conclusion should be containing one sentence on the future recommendation.

There are very long sentences which must be revised and clarify for readers understanding.

Also discuss some main findings and conclusion based on the review.

Introduction

Line 51-52, citation needed.

Lie 57-5, poor English.

Line 59-60 must be cited.

Line 66-67, what type of treatment? Need clarification with previous paragraph.

In Line 77-78 and 87-88: Natural products encompass compounds derived from plants, fungi, and bacteria must be cited the following references may be helpful.

Up to date, native people of the developing countries rely on the use of medicinal herbs to manage their healthcare issues.

The following articles can be cited;

10.30848/PJB2022-3(19), and 10.1016/j.chnaes.2020.12.006.

 The authors should mention some modern computational techniques in the introduction. Briefly.

Section 2. Line 102-103, the author states that “This medicinal herb has been traditionally used by native people in China to alleviate pain, manage inflammation and treat oesophageal cancer. This  statement need proper citations.

Line 300- 301, need reference.

In section 10, line 410-411, some recent studies about medicinal plants must be cited. i.e., 10.14719/pst.2019.6.4.571; 10.1590/0001-3765202020190387; 

The authors should mention some prominent drugs or compounds, its uses against cancer, a table would be better.

This manuscript can be accepted after recommended revision.

Reviewer 2 Report

The review deals with  studies on innovative therapeutic approach  for osteosarcoma treatments ,particularly  concerning the differentiation of tumeroids processes that can involve primary or methastatic expressions  affecting drug-unresponsive patients. I believe that by exploiting this important aspect and making efforts in experimenting different classes of natural and newly synthesised molecules inspired by active substances of plant extracts certainly worth a deeper consideration. By seeking more biocompatible remedies, we don’t expect to look for the holy-graal solutions for one of the harsher and most painful type of cancer, such as osteosarcoma of primary or secondary nature, but to discover appropriate therapies for some of the most insidious disease, which may replicate or reborn out even after a long-time period from the first occurrence in line with  the unpredictable migration of cancer cells. This renewed, but ancient approach can encounter the requirements of patients under antineoplastic treatments, but exhibiting multidrug resistance as well as being directed to fragile people with ultra-responsive immunity system. Further, these research activities are aimed to identify markers for an early identification of drug unresponsive osteosarcoma patients and to validate new antitumour agents that may be considered for novel tailored treatments, aligned  with the more recent epigenetics discoveries and achievements.

I would suggest to add some recent papers from the University of Bologna signed by Profs Boga, Calonghi and Farruggia, which utilize different natural molecules derives from sunflower seeds (racemic or enantiomeric pure HSA of different lenghts  as well as poeonia root extracts).

Round 2

Reviewer 1 Report

No further comments